# High-Fat Diet Consumption Induces Neurobehavioral Abnormalities and Neuronal Morphological Alterations Accompanied by Excessive Microglial Activation in the Medial Prefrontal Cortex in Adolescent Mice

**DOI:** 10.3390/ijms24119394

**Published:** 2023-05-28

**Authors:** Conghui Wang, Hong Li, Chen Chen, Xiuting Yao, Chenxi Yang, Zhehao Yu, Jiayi Ren, Yue Ming, Yi Huang, Yi Rong, Yu Ma, Lijie Liu

**Affiliations:** 1Medical College, Southeast University, Nanjing 210009, China; 2School of Life Science and Technology, Southeast University, Nanjing 210009, China; 3Jiangsu Provincial Key Laboratory of Critical Care Medicine, Department of Physiology, School of Medicine, Southeast University, Nanjing 210009, China

**Keywords:** high-fat diet, neuroplasticity, microglia, adolescence, mental disorders, mPFC

## Abstract

The association between a high-fat diet (HFD) consumption and emotional/cognitive disorders is widely documented. One distinctive feature of the prefrontal cortex (PFC), a kernel emotion- and cognition-related brain region, is its protracted adolescent maturation, which makes it highly vulnerable to the detrimental effects of environmental factors during adolescence. Disruption of the PFC structure and function is linked to emotional/cognitive disorders, especially those that emerge in late adolescence. A HFD consumption is common among adolescents, yet its potential effects on PFC-related neurobehavior in late adolescence and any related underlying mechanisms are yet to be established. In the present study, adolescent (postnatal days 28–56) male C57BL/6J mice were fed a control diet (CD) or a HFD and underwent behavioral tests in addition to Golgi staining and immunofluorescence targeting of the medial PFC (mPFC). The HFD-fed adolescent mice exhibited anxiety- and depression-like behavior and abnormal mPFC pyramidal neuronal morphology accompanied by alterations in microglial morphology indicative of a heightened state of activation and increased microglial PSD95^+^ inclusions signifying excessive phagocytosis of the synaptic material in the mPFC. These findings offer novel insights into the neurobehavioral effects due to adolescent HFD consumption and suggest a contributing role in microglial dysfunction and prefrontal neuroplasticity deficits for HFD-associated mood disorders in adolescents.

## 1. Introduction

Adolescence, the developmental epoch during which children become adults intellectually, physically, hormonally, and socially, is characterized by the massive remodeling of neural circuits that leads to a window of susceptibility to environmental stimuli associated with mental disturbances [1,2,3]. The first onset of numerous mental disorders usually occurs before the age of 24 [4]. Taking major depression as an example, lifetime prevalence is estimated to increase dramatically from 1% of the population under 12 years of age to ~17–25% by the end of adolescence [5]. Moreover, older adolescents with mental disorders generally have more severe episodes than their younger counterparts [6]. All of the above clearly indicate that adolescence is a unique period of vulnerability for mental health problems. Mental disorders (including depression, anxiety, and cognitive impairment) that start in adolescence not only impair academic advancement and the attainment of important developmental milestones (e.g., healthy autonomy and independence), but also lead to an elevated risk of suicide [4,6]. The consequences of mental illness in adolescents often extend to adulthood, leading to long-term morbidity and placing a substantial burden on society [4,5,6]. Therefore, the identification of significant risk factors for adolescent mental disorders and the clarification of the biological mechanisms underlying an increased risk is of great significance.

Several epidemiological and animal studies suggested that a high-fat diet (HFD) consumption adversely affects emotional and cognitive functions, including anxiety- and depression-like behaviors and learning/memory impairment [2,7,8]. Our group previously demonstrated that mice that were fed an HFD from adolescence onwards displayed greater neurobehavioral disruption than those that were administered an HFD starting from young adulthood [9], suggesting an age-related difference in the detrimental effects of prolonged HFD exposure on the brain. Due to the rapid development of the global economy and the resultant overwhelming material abundance, the overconsumption of a palatable HFD is prevalent among adolescents [2,10,11]. The negative effect of an HFD on brain function was reported to be significantly more pronounced in adolescent mice compared to adult mice [12]. However, further in-depth studies are essential to understand the detrimental impacts of HFD consumption during adolescence on cognitive/emotional behavior and the related biological mechanisms.

While the neurobiological mechanisms underlying affective–cognitive deficits are far from fully elucidated, the atrophy of the prefrontal cortex (PFC), a brain region critical for emotional regulation and cognitive control, is frequently detected in patients with dementia, depression, and other psychiatric disorders, indicating a core role of prefrontal neuroplasticity in the pathophysiology and etiology of emotional and cognitive dysfunction [13,14,15]. The PFC is one of the last brain regions to mature, with development continuing throughout adolescence and into early adulthood [16]. As a result of this delayed maturation, this region remains susceptible to environmental factors that can impair its development with lifelong consequences [16]. The studies on animal models have provided valuable insights into the potential brain regions implicated in the neuropathology of mental disorders. The disruption of the PFC structure and function may lead to behavioral changes, ranging from working memory impairment to anxiety and depression [17,18,19]. In a rodent model, the medial region of the prefrontal cortex (mPFC) was shown to be specifically related to emotional processing and cognitive function [20]. Alterations in the pyramidal neuronal morphology (including reductions in the dendritic complexity and a decreased dendritic spine density) in the mPFC might be involved in the pathophysiology of anxiety and depression [21,22]. The adult rats that were fed an HFD exhibited a reduced dendritic spine density on the pyramidal neurons of the mPFC [23], indicating that this region was a target of HFD-induced structural alterations. In view of these collective findings, we speculate that mPFC pyramidal neurons in adolescents may be susceptible to HFD consumption, and alterations in the neuroplasticity of the mPFC may be involved in the negative impact of adolescent HFD consumption on cognitive/emotional behavior.

Microglia, which are resident immune cells of the central nervous system (CNS), have emerged as a critical player in brain development, homeostasis maintenance, and neuronal circuit remodeling [24,25]. These highly dynamic cells constantly monitor the alterations in the brain microenvironment and assume different states of activation to orchestrate the context-dependent processes contributing to neuroplasticity [26]. In a healthy brain, microglia play a vital role in synaptic pruning via phagocytosis of redundant synapses, a process that is required for neuronal circuit maturation [27]. Upon stimulation, these cells undergo a series of morphological and functional changes to drive progressive neuronal injury or exert beneficial neuroprotective effects [28,29]. However, hyperactivated microglia exhibit an increased engulfment of synapses, which in turn, induce adverse effects on neuroplasticity, leading to disease initiation and progression [30,31]. Postmortem studies have revealed an excessive activation of microglia in the mPFC of patients with psychiatric disorders, implicating an aberrant microglial activity in the pathophysiology of neuropsychiatric diseases [32]. A stress-induced hyperactivation of mPFC microglia, evidenced by changes in the cell density and morphology, was associated with anxiety- and depression-like behavior [33]. Previously, our group demonstrated an association between microglial alterations and the detrimental effects of HFD consumption in adulthood that affected behavior and hippocampal neuroplasticity [25]. Further exploration of the mechanisms underlying the impact of HFD consumption during adolescence on neurobehavior and mPFC neuroplasticity, with a particular focus on microglia, is essential.

The primary objective of the present study was to evaluate the effects of adolescent HFD consumption on neurobehavior, pyramidal neuronal morphology, and the microglial status in the mPFC. Our results indicated that HFD consumption during adolescence impaired emotional behavior and mPFC neuroplasticity, which appeared to be related to a more activated microglial phenotype accompanied by increased microglial phagocytosis of the synaptic material in the mPFC. These data provide preliminary evidence for the involvement of microglia-associated prefrontal neuroplasticity deficits in HFD-induced mood disorders.

## 2. Results

### 2.1. Effects of a HFD on Food and Energy Intake and Body Weight

The food intake, caloric intake, and body weight were monitored throughout the dietary treatment period (Figure 1). As shown in Figure 2, the food intake was similar between the groups during the 4 weeks of dietary exposure (Figure 2A), but the caloric intake was significantly higher in the HFD mice relative to the control diet (CD) mice (Figure 2B, *p* = 0.0003). While the initial body weights were comparable between the groups, the animals from the HFD group exhibited significantly higher body weights than those from the CD group after one week of dietary treatment (Figure 2C, *p* = 0.0003) and until the end of the study.

### 2.2. HFD-Fed Adolescent Mice Develop Anxiety- and Depression-Like Behavior

A series of behavioral tests were performed to explore the effects of adolescent HFD consumption on emotional and cognitive functions (Figure 1). In the open field test (OFT), the central area duration (Figure 3B, Mann–Whitney test: *p* = 0.0100) and the proportion of the distance traveled in the central area (Appendix A, Mann–Whitney test: *p* = 0.0054) decreased in the HFD group, whereas the total distance of travel (Figure 3A), the traveling speed (Appendix A), and the central area entries (Figure 3C) were not affected. In the elevated zero maze (EZM) test, no statistical differences between the groups were observed in the total distance moved (Figure 3D), the open arms duration (Figure 3E), or the open arms entries (Figure 3F). These observations indicated that adolescent HFD consumption increased the anxiety-like behavior in mice but did not impair locomotor activity. Moreover, the HFD-fed mice displayed a markedly decreased sucrose preference (Figure 3H, Mann–Whitney test: *p* = 0.0103) but a normal total fluid intake (Figure 3G) during the sucrose preference test (SPT), indicative of anhedonia and depression-like behavior. However, no significant differences in the immobility time during the forced swim test (FST) were observed between the two groups (Figure 3I). To determine the effects of the adolescent HFD exposure on cognitive functions, the nest building (NB) test and temporal order recognition (TOR) task were performed to explore the ability of the animals to perform activities of daily living and demonstrate temporal order recognition memory, respectively. The nesting scores of the HFD mice were comparable to those of the CD mice at 24 h (Figure 3K) but exhibited a trend for a decrease at 12 h (Figure 3J), although this trend did not reach a statistical significance. In the TOR task, no significant difference between the groups was observed in the discrimination ratio (Figure 3L). These behavioral test results collectively suggest that HFD consumption during adolescence promoted anxiety- and depression-like behavior.

### 2.3. Adolescent HFD Consumption Alters Pyramidal Neuronal Morphology in the mPFC

To investigate the influence of the adolescent HFD exposure on prefrontal neuroplasticity, Golgi staining and a Sholl analysis were utilized for evaluating the morphology of pyramidal neurons (dendritic complexity, spine density, and spine morphology) in the prelimbic cortex (PL) and the infralimbic cortex (IL) of the mPFC. The representative images are shown in Figure 4B–C. The HFD-fed mice exhibited a decreased dendritic complexity in the mPFC, evidenced by a reduction in the dendritic length in the PL (Figure 4D, apical: *p* = 0.0008) and IL (Figure 4I, basal: *p* = 0.0466; apical: *p* = 0.0181), a reduction in the number of apical dendritic branches in the PL (Figure 4E, *p* = 0.0023), and a reduction in the number of dendritic intersections with equidistant Sholl spheres in the PL (Figure 4F) and IL (Figure 4K). These data suggest that dendritic atrophy occurred following prolonged HFD consumption during adolescence.

Next, we examined the spine density of apical dendrites where the reduction in the dendritic complexity was more pronounced. No significant difference between the groups was observed in the spine density in the mPFC (Figure 4G,L). However, a further quantitative analysis of the spine morphology disclosed that the proportion of thin (Figure 4H, PL: *p* = 0.0396) and stubby (Figure 4M, IL: *p* = 0.0044) spines was markedly increased in the HFD-fed mice, while that of mushroom spines was decreased (Figure 4M, IL: *p* = 0.0374). Taken together, the results indicated that the exposure of adolescent mice to an HFD impaired the morphological neuroplasticity of the mPFC region.

To evaluate the relationship between the emotional abnormalities and dendritic atrophy, correlations between the behavioral parameters and neuronal morphological indexes of the same mice were calculated using Pearson’s correlation test. As shown in Appendix A, the apical dendritic length in the mPFC was significantly correlated with the central area duration in the OFT (PL: *p* = 0.0014, r = 0.9158; IL: *p* = 0.0024, r = 0.8983), the open arms duration in the EZM (IL: *p* = 0.0310, r = 0.7529), and the sucrose preference in the SPT (PL: *p* = 0.0280, r = 0.7632; IL: *p* = 0.0110, r = 0.8285), suggesting that the anxiety- and depression-like behavior of the HFD-fed mice was related to the atrophy of pyramidal neurons in the mPFC.

### 2.4. HFD-Fed Adolescent Mice Show Altered Microglial Activity in the mPFC

A close correlation between the microglial activity and neuronal plasticity in neurodevelopmental and neurobehavioral abnormalities has been established [30,35]. Since morphology is significantly related to the function of microglia, we examined the microglial phenotypes in the mPFC via double staining using ionized calcium-binding adaptor molecule 1 (Iba1, a microglia-specific marker) and a cluster of differentiation 68 (CD68, a phagocytic microglial marker). The representative confocal images of the microglial cells co-stained with Iba1 and CD68 from each group are shown in Figure 5D. No statistical differences in the microglial density were observed between the two groups (Figure 5E,I). Compared to the CD group, microglia from the HFD group showed an increased soma area (Figure 5F, PL: *p* = 0.0365; Figure 5J, IL: *p* = 0.0082), but a decreased territory area (Figure 5G, PL: *p* = 0.0424; Figure 5K, IL: *p* = 0.0007), indicative of microglial hyperactivation.

In addition, the microglial phenotypic profile was characterized using scoring based on the CD68 expression within Iba1^+^ microglia on a scale ranging from zero to three, with higher scores signifying a greater microglial activation [25,36]. As shown in Figure 5L, adolescent HFD consumption led to an increase in the proportion of CD68^+^ microglia with a score of three in the IL (*p* = 0.0463), further demonstrating a higher activated state of microglia in the mPFC in response to the HFD exposure.

### 2.5. HFD Intake in Adolescence Alters Microglial Engulfment of PSD95

Abnormal phagocytosis of the synaptic material by microglia is reported to contribute to the development of neurological diseases [37]. To ascertain whether microglial phagocytosis is associated with the altered neuroplasticity observed in HFD-fed adolescent mice, z-stacks of postsynaptic density 95 (PSD95, a postsynaptic element) and Iba1 double-labeled immunofluorescence images in the PL and IL of the mPFC were acquired. We observed the microglial engulfment of the PSD95-positive puncta in all the groups of mice in three-dimensional projection images. The representative images from each group are shown in Figure 6A–F.

Similar to the data obtained on neuroplasticity via Golgi staining, the mice from the HFD group exhibited a significant decrease in the overall PSD95 density in the PL (Figure 6G1, *p* = 0.0049) and IL (Figure 6G2, *p* = 0.0021) of the mPFC. Next, we quantified the signals of the PSD95-positive puncta colocalized to Iba1 per microglia with the aid of three-dimensional surface rendering. Compared to the CD-fed mice, the volume (Figure 6H1, PL: *p* = 0.0020; Figure 6H2, IL: *p* = 0.0142) and integrated fluorescence intensity (IntDen; Figure 6I1, PL: *p* = 0.0008; Figure 6I2, IL: *p* < 0.0001) of the PSD95 puncta colocalized with Iba1 per microglia in the PL and IL were significantly increased in the HFD-fed mice. These findings supported the theory that adolescent HFD consumption promoted microglial phagocytosis of the PSD95 puncta.

### 2.6. Correlations between Behavior, Neuroplasticity, and Microglia

The associations between behavioral traits and neuroplasticity, the microglial morphological indexes, as well as microglial PSD95^+^ inclusions of the same mice were determined using a Pearson correlation analysis (only the parameters with the most significant changes were included). As shown in Appendix A, the behavioral indexes were significantly positively correlated with the PSD95 puncta density (OFT: *p* = 0.0004, r = 0.8525; SPT: *p* = 0.0240, r = 0.6433) and the microglial territory area (OFT: *p* = 0.0190, r = 0.6626; SPT: *p* = 0.0007, r = 0.8354) but negatively correlated with the PSD95 IntDen per microglia (OFT: *p* = 0.0210, r = −0.6524) in the IL, indicating that the anxiety- and depression-like behavior induced by adolescent HFD consumption was closely related to neuroplasticity, microglial morphology, and the microglial PSD95^+^ inclusions in the mPFC. Furthermore, significant correlations were found between the microglial PSD95^+^ inclusions and both the PSD95 puncta density (*p* = 0.0150, r = −0.6816) and the microglial territory area (*p* = 0.0033, r = −0.7715), indicating that impaired neuroplasticity was linked to excessive microglial phagocytosis of PSD95 in association with the microglial activity status.

## 3. Discussion

The concept that HFD consumption is associated with poor neurobehavioral outcomes is supported by an increasing body of clinical and preclinical evidence [7,8,38,39]. Adolescence is a critical time for the incidence of mental disorders associated with an increased risk of health problems, unemployment, and suicidal behavior in adulthood [40,41]. Given the global social burden imposed by adolescent emotional/cognitive disorders and the high prevalence of HFD consumption in this age group, it is important to clarify the effects and underlying biological mechanisms of HFDs on neurobehavior to provide a theoretical basis for promoting healthy dietary habits and optimizing intervention strategies for adolescents.

Consistent with previous results [2], the data from the present study showed that adolescent HFD consumption induced a decrease in the central area duration in the OFT and open arms duration in the EZM, suggesting an anxiety-like behavior. Moreover, the HFD-fed mice displayed a decreased sucrose preference in the SPT, indicative of anhedonia, a core symptom of depression in humans [42]. However, conflicting reports on the effects of HFDs on cognitive functions were documented in the literature [16,43,44,45], which could be attributable to the complex mechanisms and differences among the species, genders, ages, behavioral tests, and, in particular, feeding paradigms, such as the ingredients within the diet and the duration of consumption. In our experiments, the nesting scores used to evaluate the cognitive ability exhibited a trend for a decrease in the HFD mice at 12 h, and this difference between the two groups was attenuated at 24 h. This finding suggested that extending the observation time could result in a failure to observe cognitive differences between the groups. Our findings were similar to those of Gimenez-Llort et al. [46], who showed that transgenic mice with dementia had significantly lower nesting scores than the control mice at 48 h, but all the mice built complete dome-shaped nests at 72 h. Therefore, in future studies, the inclusion of additional observation timepoints within the 12 h period of the NB test may reveal significant effects of HFD consumption on cognitive functions.

Adolescence is a vulnerable time in terms of the susceptibility to mental disorders, which may be associated with a delayed maturation of the PFC, a key brain region correlated with emotion and cognition, during this period [16,47]. Prefrontal atrophy is a frequently reported structural neuroimaging manifestation in individuals with depression and dementia [14,48]. The studies on rodent models extend human data and confirm that neuroplasticity deficits, such as a decreased dendritic complexity and synapse density, are major causes of prefrontal atrophy and dysfunction [49]. These collective findings clearly supported prefrontal neuroplasticity having a central role in the pathophysiology of mental disorders. In adult animals, the PFC was reported to be susceptible to HFD-induced morphological and functional alterations [23]. The results of our morphological analysis of pyramidal neurons targeting two subregions of the mPFC, the PL and IL, extended the previous findings, indicating that adolescent HFD exposure induced dendritic atrophy of the target regions, evidenced by a reduction in the dendritic length, the number of apical dendritic branches, and the number of dendritic intersections with equidistant Sholl spheres. Furthermore, the Pearson correlation analysis revealed significant associations between the apical dendritic length and behavior parameters (the central area duration in the OFT, open arms duration in the EZM, and sucrose preference in the SPT), indicating that anxiety- and depression-like behavior induced by adolescent HFD consumption was closely linked to neuronal morphology in the mPFC. Dendritic atrophy may result in the dysregulation of information transfer and processing, thereby inducing emotional abnormalities [50].

Dendritic spines are tiny membranous protrusions on the surface of neurons and constitute over 90% of the excitatory synapses in the CNS [51]. Changes in the dendritic spine number, size, and shape are associated with the plasticity of synapses, as well as emotional and cognitive impairment [52]. The morphology of the spines is highly variable and can usually be classified into three types, specifically, mushroom, thin, and stubby [53]. With maturation, the spines tend to stabilize [54]. Mushroom spines with large heads are considered mature and functional and contribute to strong, long-term synaptic connections [53,55]. Thin and stubby spines are considered immature and unstable, leading to weak or silent synaptic connections [53,55]. The mice with depression-like symptoms exhibited alterations in the spine morphology in the mPFC, such as a reduction in the proportion of mushroom spines and an increase in the number of thin spines [56]. In keeping with these results, the proportion of mushroom spines in the mPFC decreased, while that of thin and stubby spines increased after HFD consumption by adolescent mice in our study. The different spine types have distinct functions and changes in their proportions may have a significant impact on neuronal excitability and function, eventually leading to an increased incidence of neurobehavioral deficits [27,53]. In conclusion, we demonstrated a significant adverse effect of HFD feeding during adolescence on the pyramidal neuronal morphology in the mPFC for the first time, as evident from the decrease in the neuronal complexity and mature dendritic spines and the increase in the immature spines. The HFD-induced abnormal neuronal morphology of the mPFC may serve as the pathological mechanism underlying anxiety- and depression-like behaviors in our paradigm.

While emotional disorders such as anxiety and depression have long been considered to be of neuronal origin, microglial dysfunction has recently been robustly implicated in the onset and development of these disorders, given their high sensitivity to environmental factors and pivotal role in shaping neuroplasticity [57]. Several recent studies have shown that a HFD induces cognitive impairment accompanied by microglial activation in the PFC, supporting the involvement of microglia in HFD-induced behavioral dysfunction [43]. Microglia act as the information sensors for changes in the CNS microenvironment and play a crucial role in maintaining homeostasis and regulating neuroplasticity, in turn affecting emotional and cognitive behavior [25,58]. These key cells continuously monitor the local environment, engulf cellular debris, and prune inappropriate synapses to maintain optimal neural networks [59,60,61]. Disorders of homeostasis in the brain can cause rapid and long-lasting alterations in the morphology and function of microglia [28]. Hyperactivated microglia typically exhibiting enhanced phagocytic activity are commonly detected in the PFC of patients with psychiatric disorders and stress-induced animal models of depression [30,32]. Consistent with this finding, in our study, microglia displayed a more activated phenotype in the mPFC of the HFD-fed adolescent mice, characterized by a significant decrease in the territory area and an increase in the soma area, as well as the CD68 expression. In addition, we found strong correlations between the microglial territory area with both anxiety- and depression-like behaviors (indicated by the central area duration in the OFT and the sucrose preference in the SPT) and the synapse density in the mPFC. These observations led us to hypothesize that an altered microglial activity induced by HFD exposure in adolescence may impact mPFC neuroplasticity, culminating in the impairment of neurobehavior. 

During adolescence, the microglia activity-dependent elimination of synapses serves to modulate the maturation of neuronal synapses and improve the efficiency of the brain network [30,47]. However, aberrant microglial phagocytosis of the synaptic material is linked to the pathophysiology of neuropsychiatric diseases [62]. Consistent with our Golgi staining data showing an adverse effect of adolescent HFD exposure on prefrontal neuroplasticity, the quantitative analysis of the fluorescent signal of PSD95, a postsynaptic marker, revealed that HFD consumption during adolescence induced a significant decrease in the overall synapse density in the mPFC. Fluorescence double labeling with PSD95 and Iba1 revealed a marked increase in the volume and integrated fluorescence intensity of the engulfed PSD95 puncta per microglia in the mPFC of the HFD-fed adolescent mice. The correlation analysis further revealed strong associations between the microglial territory area and the microglial PSD95^+^ inclusions, between the microglial PSD95^+^ inclusions and the synapse density, and between the synapse density and anxiety- and depression-like behavior. Combined with our observations of alterations in microglial CD68 expression (a lysosomal marker indicative of the phagocytic activity of microglia), these data indicated that the microglial engulfment of synaptic material that is related to microglial activity status was increased. Based on these collective findings, we speculated that adolescent HFD-induced prefrontal neuroplasticity impairment and mood disorders may be attributable to excessive microglial phagocytosis of the synaptic elements in the mPFC.

The altered microglial activity induced by a HFD interferes with the physiological crosstalk between microglia and the synapses, and the abnormal synaptic pruning resulting from aberrant microglial activity can lead to an abnormal neuronal morphology [63]. Moreover, other preclinical studies have demonstrated that HFD consumption results in metabolic disorders, gut microbiota dysbiosis, excessive activation of microglia leading to neuroinflammation, and, ultimately, induces anxiety- and depression-like behavior [64,65,66]. The modulation of the microglial activity plays a critical role in influencing neuroplasticity in key mood-regulating regions, such as the mPFC. It should be noted that the data presented here are correlational, and thus a causative direction cannot be drawn. However, considering the essential interdependence that exists between microglia and neurons, it is reasonable to hypothesize that microglia disturbed by HFD consumption are impaired in their abilities to properly support neuroplasticity, and hence may create or fuel a vicious cycle between microglial dysfunction and the neuroplasticity deficits, driving further progression of mood disorders. Further experiments on the suppression of the microglial activation are imperative to determine the causative role of microglia in prefrontal neuroplasticity deficits and emotional disorders in adolescents with HFD exposure.

In conclusion, HFD consumption during adolescence clearly induced anxiety- and depression-like behavior in mice. We demonstrated adverse effects of adolescent HFD consumption on the mPFC pyramidal neuronal morphology for the first time, which may be linked to enhanced microglial phagocytosis of the synaptic elements in association with microglial hyperactivation in the mPFC. Our findings highlighted the importance of maintaining healthy dietary habits during sensitive windows of neurodevelopment, such as adolescence, and provide further evidence for the crucial role of prefrontal pyramidal neurons and microglia in HFD-induced neurobehavioral impairment. The data from our study collectively suggest that microglia are key mediators of neuroplasticity and behavioral abnormalities, providing novel potential therapeutic targets and intervention strategies for mood disorders.

## 4. Materials and Methods

### 4.1. Animals and Experimental Design

The male C57BL/6 mice (postnatal day 21, P21) were purchased from Changzhou Cavens Experimental Animal Co., Ltd. (Changzhou, China, SCXK (SU) 2021-0013). All the mice were housed under standard conditions (7 a.m.–7 p.m. light cycle, 22 °C, 55% humidity, and feed and water changed every four days) following the guidelines of the University Committee for Laboratory Animals of Southeast University (Nanjing, China). Following one week of adaptive feeding (P28), the mice with similar average body weights were divided into two groups and housed in cages (29 cm (length) × 22 cm (width) × 14 cm (height), four to five per cage). The animals were provided free access to either an HFD (5.0 kcal/g: 60% energy from a lard and soybean oil mixture at a ratio of ~10:1, 20.6% from carbohydrates, and 19.4% from protein; TP23300, Trophic Animal Feed High-Tech Co., Ltd., Nantong, China) or a CD (3.5 kcal/g: 10% energy from fat, 70.6% from carbohydrates, and 19.4% from protein; TP23303, Trophic Animal Feed High-Tech Co., Ltd.) for 4 weeks. During this period, their body weights were recorded weekly, the food intake of the animals from each cage was measured every four days, and the average caloric intake per animal per day was calculated. The caloric intake was determined by multiplying the food intake by the calorie content. On completion of the 4-week dietary treatment (P56), the mice underwent behavioral tests followed by tissue collection, as shown in the schematic illustration of the experimental procedure (Figure 1). The mice continued to consume their respective diets throughout the study period. The dietary treatment period was selected to cover almost the entire adolescence stage, which is the period of susceptibility to mental disorders [4,67,68]. The individual mice were identified by ear tags throughout the study. All the experimental procedures were conducted in accordance with the protocols approved by the University Committee for Laboratory Animals of Southeast University. The investigators were blinded to the dietary conditions throughout the behavioral tests, histological analyses, and quantification analysis.

### 4.2. Behavioral Tests

At P56–62, coincident with the prevailing definition of late adolescence in rodents, which is the peak age of susceptibility for mental disorder episodes [5,68], the following sequence of experiments was conducted on the mice during the light period: the NB test, SPT, OFT, TOR task, EZM test, and FST. Due to time limitations, 12 mice were randomly selected from each group for the TOR task, whereas all the mice were subjected to the other behavioral tests. To minimize the effect of the behavioral tests on the mice, all the animals were transferred to the testing room at least 2 h before the experiments for acclimatization to the test environment, and we performed the behavioral tests beginning with the least invasive and ending with the most invasive (i.e., from the least to the most stressful) [69]. All the behavioral tests, except the NB test and SPT, were recorded using a video camera and analyzed using a digital tracking system (Visutrack 3.0; Shanghai Xinruan Information Technology Company, Shanghai, China). At the end of each test, the individual mice were returned to their home cages and the apparatus was cleaned with 70% ethanol and water to remove olfactory cues.

#### 4.2.1. OFT

The OFT was used to investigate the locomotor activity and anxiety-like behavior, as described previously [2]. Briefly, the individual mice were placed in an open field arena (50 cm × 50 cm × 50 cm) facing the same wall and were allowed to freely explore for 5 min. The total distance traveled in the field, the number of entries into the delineated central zone (25 cm × 25 cm square field in the middle of the arena), and the time spent in the central zone were determined.

#### 4.2.2. EZM Test

The EZM test was employed to measure anxiety-like behavior [70]. The experimental device was composed of a pair of open arms (with 1 cm high curbs to prevent falling) and a pair of closed arms (with 20 cm walls from the surface of the maze) elevated 46 cm from the ground. The mice were placed in a closed arm and allowed to explore the space for 5 min. The total distance moved in the mazes, the number of entries into the open arms, and the time spent were determined.

#### 4.2.3. SPT

As a widely used behavior marker of anhedonia, the SPT is commonly applied to measure depression-like behavior in animals [71]. Prior to the test, the individually housed mice were habituated to two identical bottles containing water for 24 h. On the testing day, the mice were exposed to two pre-weighed bottles, one containing water and the other containing a 1% sucrose solution for 24 h. The positions of the two bottles were switched every 12 h to reduce the position preference. The sucrose and water consumption was measured after 24 h. The sucrose preference was calculated as follows: sucrose consumption/(sucrose consumption + water consumption).

#### 4.2.4. FST

The FST was conducted to assess depressive-like behavior in mice [72]. The mice were individually placed in a transparent glass cylinder (with a 13 cm diameter and 18.5 cm high) containing 15 cm of fresh water maintained at 23–25 °C. Each mouse was forced to swim for 6 min, and the duration of immobility was measured during the last 4 min of the test. The mice were considered to be motionless when floating while performing minimal necessary movements to maintain their nose above the water.

#### 4.2.5. NB Test

The NB test facilitated the evaluation of the daily living activities that are typically altered in patients with cognitive impairment [43]. For this experiment, mice were housed individually in cages containing nesting material for 24 h and were photographed every 12 h. The nest quality was scored on a five-point scale based on the study of Harauma et al. [73], with higher scores representing a better quality. A score of one signified no nest; two, a flat nest; three, a cup- or bowl-shaped nest; four, an incomplete dome-shaped nest; and five, a complete dome-shaped nest.

#### 4.2.6. TOR Task

The TOR task was used to assess the ability of an animal to differentiate between two familiar objects, presented previously at different times [74]. This task comprised two sample phases and one test trial. In each sample phase, the animals were allowed to explore two identical objects for a total of 5 min. Different objects were used for sample phases I and II, with a 1 h delay between the phases. The test trial (5 min duration) was performed 3 h after sample phase II. During the trial, one object from sample phase I and one from sample phase II were provided, and the time spent exploring each object was calculated. Object exploration was defined as active contact with the object via the nose, foreclaws, or whiskers [75]. According to this definition, a mouse standing near an object without interacting with it would not be classified as object exploration. In cases where the temporal order memory is intact, animals generally spend more time exploring the object from sample I (i.e., the novel object presented less recently) compared to that from sample II (i.e., the familiar object presented more recently) in the test trial. The discrimination ratio (DR) was calculated as follows: (exploration of novel object − exploration of familiar object)/(exploration of novel object + exploration of familiar object).

### 4.3. Tissue Collection

The day after the FST, the mice were weighed and injected (i.p.) with pentobarbital (100 mg/kg). For Golgi staining, four brain samples per group were maintained in a mixture of solutions A and B from the FD Rapid GolgiStain^TM^ Kit (FD Neuro Technologies., Inc., Columbia, MD, USA) overnight at room temperature and away from light, with the solutions being subsequently replaced with fresh solutions A and B and being incubated for 14 days in the same environment. Next, the brain samples were transferred to solution C from the FD Rapid GolgiStain^TM^ Kit for 7 days at room temperature and processed according to the manufacturer’s instructions [76,77]. For the immunohistochemical analysis, eight animals per group were perfused transcardially with 20 mL of 0.9% saline followed by 20 mL of 4% paraformaldehyde (PFA) in a 0.1 M phosphate-buffered saline (PBS). The brains of the mice were rapidly excised, fixed in 4% PFA, cryoprotected in 30% sucrose in PBS until the tissue sank in the solution, and embedded in an optimal cutting temperature (OCT) compound before cryosectioning. According to Paxinos and Franklin’s mouse brain atlas [34], serial coronal sections (40 μm thick) of the mPFC (bregma of 1.98–1.54 mm) were obtained.

### 4.4. Golgi–Cox Impregnation and Morphometric Analyses

Coronal sections (150 µm thick) of the Golgi–Cox-impregnated brain samples were obtained using a vibratome (Leica VT 1200S, Nussloch, Germany). The sections were mounted on gelatin-coated slides, air-dried, dehydrated in graded ethanol, and cleared in xylene. All the images were acquired via light microscopy (Olympus BX53, Tokyo, Japan). The PL and IL of the mPFC (bregma of 1.98–1.54 mm) were identified based on Paxinos and Franklin’s mouse brain atlas [34]. The well-impregnated, individual pyramidal neurons in the PL and IL regions were traced and measured using the NeuronJ plugin of the ImageJ software 1.52a (US National Institutes of Health, Bethesda, MD, USA). The Sholl analysis was performed to analyze the dendritic length, the number of dendritic branches, and the number of dendritic intersections with concentric circles at radial intervals of 10 µm. The dendritic spines were quantified on secondary or tertiary apical branches located about 10 μm from the bifurcation point (at least 15 µm in length) and expressed as the number of spines per a dendrite length of 10 μm. According to the previous reports [53], the spines could be distinguished into three categories: (1) mushroom spines with smaller neck diameters than head diameters, (2) thin spines with greater lengths than diameters and similar neck and head diameters, and (3) stubby spines with lengths similar to their diameters. The proportion of each type was calculated as follows: spine number of each type on an individual branch/total spine number on an individual branch.

### 4.5. Immunohistochemistry

Staining was performed using the selected free-floating sections. Two to four sections at 80 μm intervals were selected from each mouse sample, washed three times with 0.1 M of PBS, and blocked with a blocking solution for 1 h at room temperature. Next, the sections were incubated with the following primary antibodies overnight at 4 °C: rabbit anti-Iba1 (for microglia, 1:1000, 019–19741; Wako, Osaka, Japan), rat anti-CD68 (for phagocytic microglia, 1:1000, MCA1957; Bio-Rad, Oxford, UK), goat anti-Iba1 (1:600, 011–27991; Wako), and rabbit anti-PSD95 (for the postsynaptic element, 1:1000, ab18258; Abcam, Cambridge, UK). After washing three times with PBS, the sections were incubated with the following secondary antibodies for 2 h at room temperature away from light: Alexa-568 goat anti-rabbit (1:1000, ab175471; Abcam), Alexa-488 goat anti-rat (1:1000, A11006; Thermo Fisher Scientific, Waltham, MA, USA), Alexa-488 donkey anti-goat (1:1000, ab150129; Abcam), and Alexa-594 donkey anti-rabbit (1:1000, ab150064; Abcam). All the slides were counterstained in PBS with 4′6-diamidino-2-phenylindole (DAPI, 1:600, C1027; Beyotime, Shanghai, China) at room temperature for 15 min to visualize the cell nuclei. The negative control sections were incubated without primary or secondary antibodies.

### 4.6. Quantitative Analysis of Immunohistological Data

The images were obtained using a confocal microscope (Olympus FV1000 or Olympus FV3000, Tokyo, Japan) under 40× and 60× objectives. The confocal z-stacks were acquired at a resolution of 1024 × 1024 pixels with a 1 µm step size. The PL and IL regions were identified according to Paxinos and Franklin’s mouse brain atlas [34]. The samples were analyzed by a trained observer blinded to the experimental group using the ImageJ software 1.52a or the Imaris software 9.8.0 (Bitplane, Oxford, UK). 

For the evaluation of microglia, only the Iba1^+^ cells with a clearly visible cell body were analyzed in the PL and IL regions. The quantitative analysis was conducted using the microglial phenotypic profile with the following parameters [2]: (1) the microglial density, defined as the number of Iba1^+^ cells per area; (2) the average microglial soma area; (3) the average microglial territory area, defined as the two-dimensional area formed by connecting the outermost points of the Iba1^+^ cell dendritic processes; and (4) the percentage of CD68^+^ microglia scoring zero to three (defined as the proportion of CD68^+^/Iba1^+^ cells with scores of zero to three among all the Iba1^+^ cells). As reported previously, CD68 expression was scored as zero (no/scarce expression), one (punctate expression), two (expression covering one-third to two-thirds of the entire Iba1^+^ cell soma area), or three (expression covering more than two-thirds of the area) [36,78]. Higher scores were representative of a higher phagocytic activity of microglia.

For the quantitative evaluation of the microglial phagocytosis of the synaptic material, the PSD95^+^ puncta and Iba1^+^ cells in the PL and IL were analyzed by applying a three-dimensional surface rendering of the confocal stacks in their respective channels using the Imaris software. To ensure that only the PSD95 puncta entirely engulfed by microglia were included for analysis, a new channel for “engulfed PSD95” was created by masking the PSD95 signal within the Iba1^+^ signal using the mask function of Imaris [79,80]. Three to six individual microglia were analyzed per image. The volume and integrated intensity of the PSD95 puncta colocalized with Iba1 per microglia in the PL and IL regions of the mice from the HFD and CD groups were measured to assess the microglial phagocytosis of PSD95. Overall, the PSD95 puncta density was expressed as the number of PSD95 puncta per 1000 µm^3^ in the brain region.

### 4.7. Statistical Analysis

The statistical analysis and graph generation of the experimental data were performed using the GraphPad Prism 8.0 (GraphPad Software Inc., San Diego, CA, USA) and Origin software (Version 2022, Origin Lab Corp., Northampton, MA, USA). The statistical differences were determined using the Mann–Whitney test or the two-tailed unpaired Student’s *t*-test. The tests used for the analysis of each experiment are described in the figure legend and/or in the Results section describing the experiments. Pearson’s correlation coefficient analysis was used to assess the relationship between the behavioral parameters and the neuronal or microglial morphological parameters of the same mice. All the values were expressed as the mean ± the standard error mean (SEM). The data were considered statistically significant at *p* < 0.05.

## Figures and Tables

**Figure 1 ijms-24-09394-f001:**
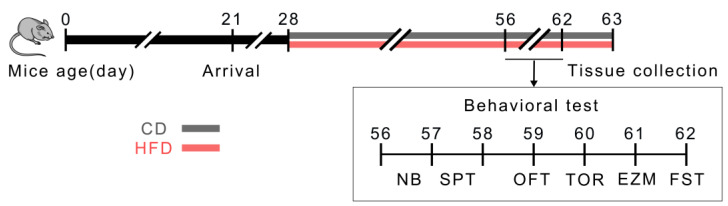
Schematic illustration of the experimental procedure and behavioral tests. CD: control diet; HFD: high-fat diet; NB: nest building; SPT: sucrose preference test; OFT: open field test; TOR: temporal order recognition; EZM: elevated zero maze; FST: forced swim test.

**Figure 2 ijms-24-09394-f002:**
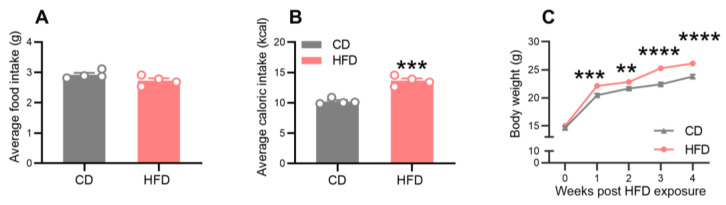
Effects of adolescent HFD consumption on the food intake, caloric intake, and body weight. (**A**) Average daily food intake per mouse. (**B**) Average daily caloric intake per mouse. (**C**) Weight changes in the mice during the 4-week dietary intervention period. Values are presented as the mean ± the SEM. *n* = four cages per group (four to five mice per cage) in (**A**,**B**). *n* = 17–18 mice per group in (**C**). ** *p* < 0.01, *** *p* < 0.001, and **** *p* < 0.0001 compared to the CD group via the Student’s *t*-test. Gray and red circle points/bars represent CD- and HFD-fed mice, respectively.

**Figure 3 ijms-24-09394-f003:**
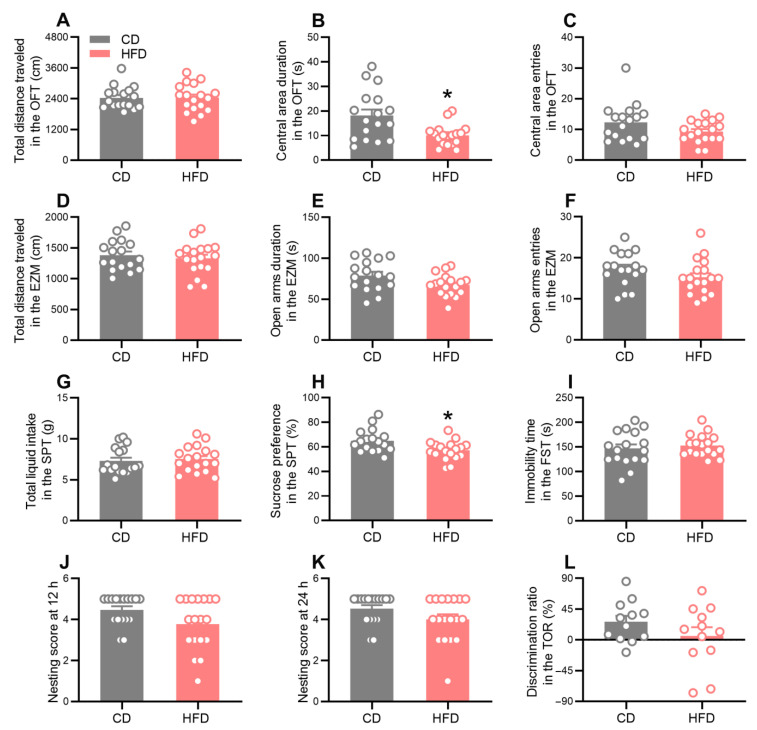
Effects of adolescent HFD consumption on neurobehavior in mice. (**A**–**C**) Total distance traveled (**A**), central area duration (**B**), and central area entries (**C**) in the OFT. (**D**–**F**) Total distance traveled (**D**), open arms duration (**E**), and open arms entries (**F**) in the EZM. (**G**,**H**) Total liquid intake (**G**) and sucrose preference (**H**) in the SPT. (**I**) Immobility time in the FST. (**J**,**K**) Nesting score at 12 h (**J**) and 24 h (**K**) in the NB test. (**L**) Discrimination ratio in the TOR task. Values are presented as the mean ± the SEM. *n* = 17–18 mice per group in (**A**–**K**). *n* = 12 mice per group in (**L**). * *p* < 0.05 compared to the CD group via the Mann–Whitney test. Gray and red circle points/bars represent CD- and HFD-fed mice, respectively. Each data point represents an individual mouse.

**Figure 4 ijms-24-09394-f004:**
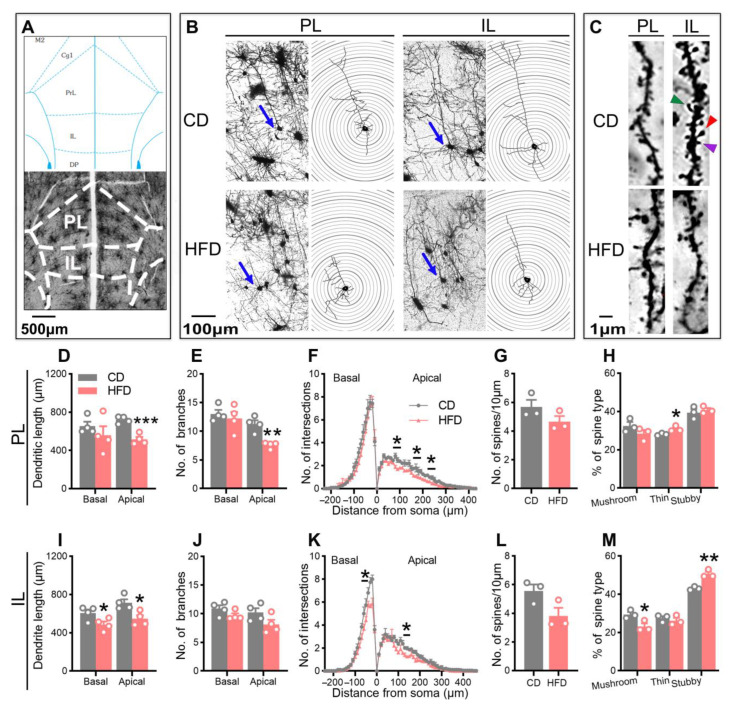
Effect of adolescent HFD consumption on the morphology of pyramidal neurons in the medial prefrontal cortex (mPFC). (**A**) Top: plane of interest of Paxinos and Franklin’s mouse brain atlas (bregma of 1.70 mm) [34]. Bottom: image of a coronal brain section with Golgi staining, with the corresponding mask of the mouse brain atlas presented on the top. (**B**) Representative traces of pyramidal cells in the prelimbic cortex (PL) and infralimbic cortex (IL) of the mPFC, along with a graphic description of the Sholl analysis parameters for the cells indicated by the blue arrows. (**C**) Representative photomicrographs of the secondary apical branches of pyramidal neurons in the PL and IL regions. Red, green, and purple arrows indicate the mushroom, thin, and stubby spines, respectively. Determination of the dendritic length (A1–A2) (**D**,**I**), number of branches (**E**,**J**), and number of intersections (Sholl analysis) (**F**,**K**) of pyramidal cells in the PL and IL. Measurement of the number of dendritic spines/10 µm (**G**,**L**) and proportion of the spine subtypes (**H**,**M**) in the PL and IL. Values are presented as the mean ± the SEM. *n* = four mice (40–46 neurons) per group in (**D**–**F**,**I**–**K**). *n* = three mice (38–42 spines) per group in (**G**,**H**,**L**,**M**). * *p* < 0.05, ** *p* < 0.01, and *** *p* < 0.001 compared to the CD group via the Student’s *t*-test. Gray and red circle points/bars represent CD- and HFD-fed mice, respectively. Each data point represents an individual mouse.

**Figure 5 ijms-24-09394-f005:**
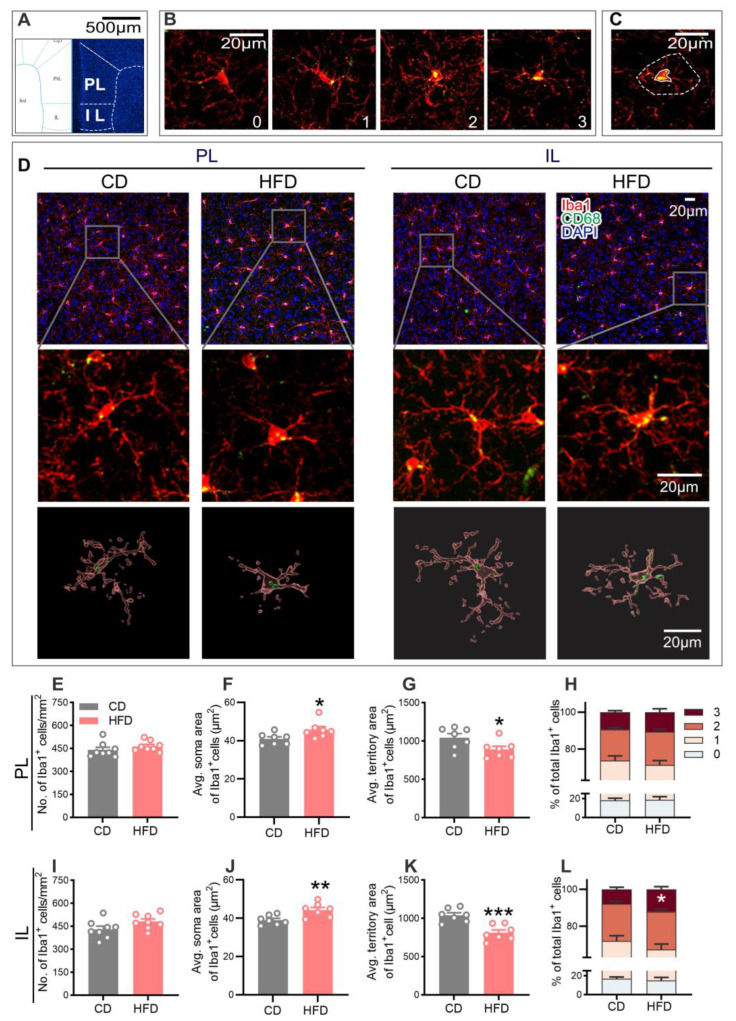
Effect of adolescent HFD consumption on the morphology of microglia in the mPFC. (**A**) Left: plane of interest of Paxinos and Franklin’s mouse brain atlas (bregma of 1.78 mm) [34]. Right: image of a coronal brain section with DAPI staining, with the corresponding mask of the mouse brain atlas presented on the left. (**B**) Representative images of the scoring of CD68 occupancy within the Iba1^+^ cells. (**C**) Representative images of the soma area (solid lines) and territory area (dashed lines) of microglia. (**D**) Representative images of Iba1 (red), CD68 (green), and DAPI (blue) immunofluorescence, and the three-dimensional reconstruction of microglia (red) and CD68 (green) within microglia using the Imaris software 9.8.0 in the PL and IL regions of the CD- and HFD-fed mice. Microglial density (**E**,**I**), average microglial soma area (**F**,**J**), average microglial territory area (**G**,**K**), and percentage of CD68^+^ microglia with scores from zero to three (**H**,**L**) in the PL and IL of the mPFC. Values are presented as the mean ± the SEM. *n* = seven to eight mice per group. * *p* < 0.05, ** *p* < 0.01, and *** *p* < 0.001 compared to the CD group via the Student’s t-test. Gray and red circle points/bars represent CD- and HFD-fed mice in (**E**–**G**,**I**–**K**), respectively. Different colored bars represent CD68^+^ microglia with scores from zero to three in (**H**,**L**). Each data point represents an individual mouse.

**Figure 6 ijms-24-09394-f006:**
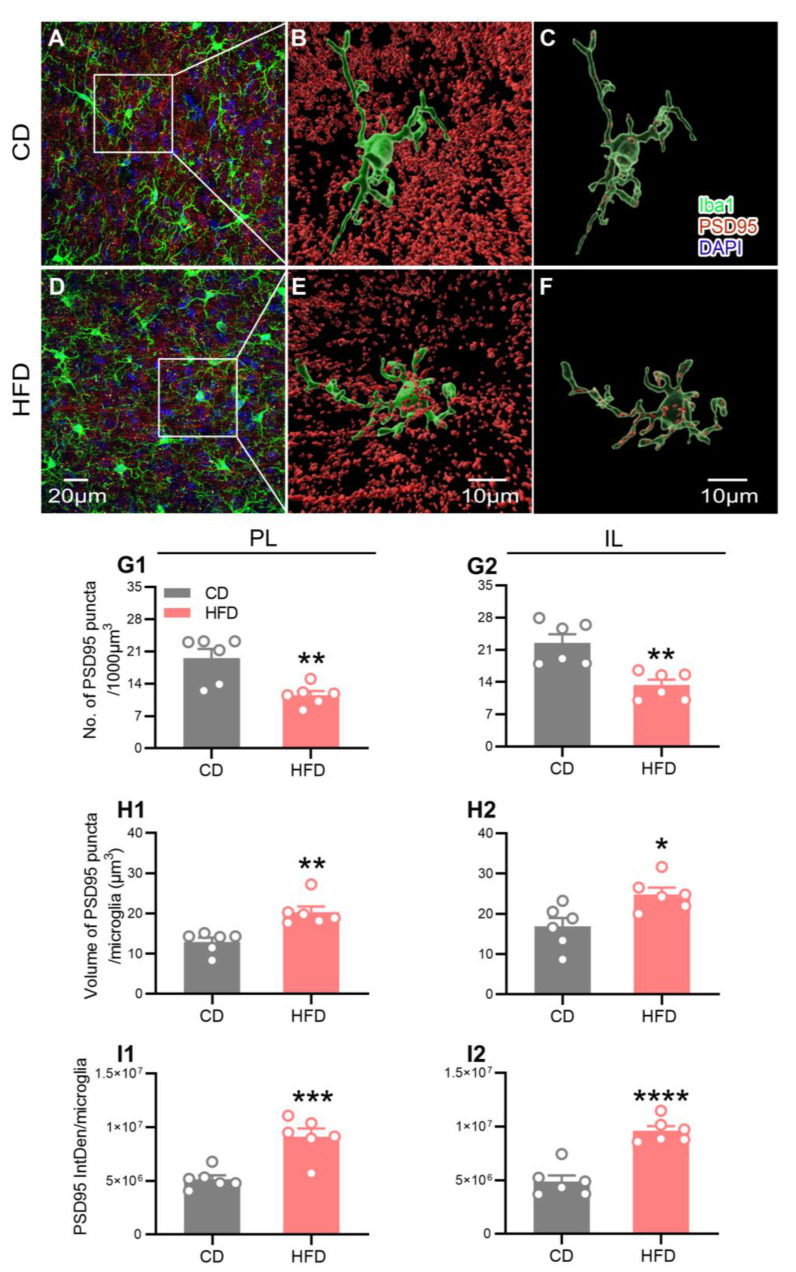
Effect of adolescent HFD consumption on the microglial phagocytosis of the PSD95 puncta in the mPFC. (**A**,**D**) Representative images of Iba1 (green), PSD95 (red), and DAPI (blue) immunofluorescence labeling in the IL of the CD- and HFD-fed mice. Three-dimensional reconstruction of microglia (green) and the PSD95 puncta (red) outside (**B**,**E**) or within (**C**,**F**) microglia using the Imaris software 9.8.0. (**G1**,**G2**) Density of the PSD95-positive puncta in the PL and IL. Volume (**H1**,**H2**) and integrated fluorescence intensity (IntDen) (**I1**,**I2**) of PSD95 puncta colocalized with Iba1 per microglia in the PL and IL. Values are presented as the mean ± the SEM. *n* = six mice per group. * *p* < 0.05, ** *p* < 0.01, *** *p* < 0.001, and **** *p* < 0.0001 compared to the CD group via the Student’s *t*-test. Gray and red circle points/bars represent CD- and HFD-fed mice, respectively. Each data point represents an individual mouse.

## Data Availability

Data contained within the paper are available from the authors upon reasonable request.

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
