# Peer review of "High-Fat Diet Consumption Induces Neurobehavioral Abnormalities and Neuronal Morphological Alterations Accompanied by Excessive Microglial Activation in the Medial Prefrontal Cortex in Adolescent Mice"

_ijms, 2023, doi:10.3390/ijms24119394_

Round 1
Reviewer 1 Report (New Reviewer)
Adolescence has been denoted as a very vulnerable period accompanied by mental disorders such as anxiety and depression. Since the brain is particularly vulnerable to environmental impact during this period, the authors have investigated the impact of a high-fat diet on behavioral alterations and microglial activation in adolescent mice. This experimental study provides new and interesting data with a valid theoretical framework. It identifies and fills the knowledge on this interesting topic. It is very well designed, with a thorough methodology and results presented clearly. I suggest to the Editor that the article can be accepted in the present form.
Author Response
Thanks for your encouragement and comments. We appreciate the reviewer's time and effort.
Reviewer 2 Report (New Reviewer)
Wang et al investigate effects of high-fat diet consumption in adolescent mice on behaviors and neuronal and microglial morphology in mPFC. Topic is interesting, and experiments are well designed and executed. They carefully analyzed neuronal and microglial characteristics in mPFC with or without high-fat diet. Also, manuscript is written fine overall. However, there are significant concerns in their data analysis, which would preclude the publication of this manuscript.
While the authors perform Pearson correlation analysis to investigate correlation between behavioral differences and anatomical/histological alterations upon high-fat diet consumption, this does not add any meaningful insight to this manuscript. In my opinion, this analysis would rather confuse readers. Most critically, it is completely unclear how their correlation analysis was conducted in this study, as there is no clear description in Materials and Methods about the analysis. Were the mice used in this study individually tracked throughout the testing and analysis? It might make sense to use such analysis to investigate correlation between behavioral variability and histological variability among individual mice in each condition (though this would require so many N to test such correlation), as mouse behaviors are often very variable. However, the authors already show changes in mouse behaviors and neuronal/microglial morphology upon high-fat diet, so their correlation analysis is unnecessary. Figures 6 and 10 and corresponding text should be removed from the manuscript.
As the authors are aware, the findings in this study regarding mouse behaviors and neuronal/microglial morphology changes are correlative. Therefore, there is no firm evidence that microglia causally contribute to behavioral or neuronal changes. Changes in microglial morphology, activation status, and PSD95 inclusion could merely be secondary to neuronal changes by high-fat diet, rather than the possibility that high-fat diet is directly affecting microglial biology. The manuscript has to be revised carefully, particularly in the abstract.
Other cocerns:
– I would strongly urge authors to change most of the graphs in the manuscript so that the y-axis starts from zero. Using different scales for each graph is very confusing and appears somewhat deceptive.
– It is extremely informative to add values from individual biological replicate as dots over the bar graph, so that readers can know sample variability.
– While the authors state that data are considered statistically significant at p < 0.05 in Materials and Methods, they frequently treat changes that are p > 0.05 as biologically meaningful. It also appears that the authors arbitrarily show p values that are between 0.05 and 0.1 only when they want to claim biological changes. This significantly loses trustworthiness and validity of this study.
– Were the investigators kept blinded to the diet conditions throughout behavior tests, histological analyses, and quantification?
– Some of the figures should be combined (like Figs 4 and 5, and 7 and 8) as they are derived from the same experiments.
– Line 154-160: The authors state that dendritic atrophy occurred in response to high-fat diet, but there is no data showing that the changes are due to the decline of dendritic complexity or retarded development.
Author Response
Please see the attachment

Reviewer 3 Report (Previous Reviewer 1)
After reviewing the paper, and the recommendations I provided in the previous review, I consider that all the necessary changes have been made to the paper, improving its quality and making it publishable from my point of view.
Author Response
We sincerely appreciate the time and effort that the editors and the reviewers have dedicated to providing feedback on our manuscript and are very grateful for all insightful comments and constructive suggestions, which helped us to improve the quality of the paper.
Reviewer 4 Report (New Reviewer)
In the manuscript submitted by Wanget. al. titled as ‘High-fat diet consumption induces neurobehavioral abnormal- 2
ities and neuronal morphological alterations accompanied by excessive microglial activation in the medial prefrontal cortex in adolescent mice’, the authors found high fat diet (HFD) in adolescent mice increased anxiety, and depression behavior, and these cognitive impairments were correlated with the abnormal morphology of neurons and microglia in the prefrontal cortex, as well as the increased synaptic phagocytosis of microglia.
Overall, this paper is of certain novelty and significance to understand the detrimental roles of HFD at a young age in the development of adolescent mood disorders. The paper itself is well written and organized with clear logic. Still, I have some major and minor comments.
Major comments:
1. In the open field test, authors showed the reduction of center area duration, non-significant reduction of center area entries, and non-affected total travelling distance. However, the travelling speed was not shown. To examine the anxiety behavior, a direct index is suggested to calculate and apply, the ratios of distance traveled in the center to the total distance.
2.We commonly use 0.05 as the threshold for significance. Any numbers larger than 0.05 would be considered as nonsignificant. P values >0.05 appeared in line 131, 145, 187, 222, please make similar statement as Line 147 or Line 163 to avoid over conclusion.
3. All the bar graph are suggested to show the datapoints stratified into male and female with different colors. This style will help readers to learn the data distribution.
4. The authors used t-tests for the comparisons between the two groups for behavioral tests. Mann Whitney test is suggested to use instead since it could be applied to gaussian or non-gaussian distributed datasets.
3. Figure 5, 8 and 9, it is unclear whether the comparisons between groups take sample or cells as entity. However, please make sure the comparison is taking mouse as entity, the datapoints to be shown should represent each mouse.
Minor comments:
1. Fig. 1. The time line looks confusing since the length from Day56 to day62 is more than the length for four weeks (1-28,28-56). Please adjust the length to remove the confusion.
2. Fig. 6. The colormap looks confusing since there were no negative correlations.
Round 2
Reviewer 2 Report (New Reviewer)
Wang et al significantly improved their manuscript in responding to the comments by the reviewers. It is great that the authors took my comment about statistical significance seriously, but it is totally wrong to use multiple statistical tests to "prove" significance. We are supposed to select an appropriate statistical test based on data structure and distribution, then present statistical likelihood whether the data is different or not. Also, we need to be aware that statistical significance is just statistical significance, and it's different from biological significance. I would urge the authors to think through about what statistical test is appropriate in Figure 3. I do not mind the authors discussing trend of increase or decrease without p < 0.05 statistical significance, if they really think that such trend is biologically important and is worth mentioning.
Along the similar line, I can't accept the sentence "... Student's t-test showed that HFD-fed mice spent less time..." (line 138-). It is NOT the statistical test, but it is experiments and investigations what gives us results. This sentence needs to be revised.
Regarding Figures 5 and 8 (Pearson correlation analysis), I would still think that these data distract readers and do not add much meaningful significance to this paper. However, since the authors added these figures in response to another reviewer, I would leave it to the authors and the editor whether these figures would be removed. One middle ground compromise might be to move over these figures to the supplementary material.
I just wanted to double check if individual mice were tracked by certain identifier (ear tag, toe clip, etc) during behavior tests. The manuscript says 4-5 mice were housed together in one cage, but compares behavioral and neuronal/microglial phenotypes in the same mice. I think it is crucial to know if this is the case, as this information is not found in the manuscript.
Author Response
Please see the attachment.

This manuscript is a resubmission of an earlier submission. The following is a list of the peer review reports and author responses from that submission.
Round 1
Reviewer 1 Report
In the manuscript “High-fat diet consumption induces neurobehavioral abnormalities and neuronal morphological alterations accompanied by excessive microglial engulfment and reduced BDNF levels in the medial prefrontal cortex in adolescent mice” Conghui Wang and co-workers examined the effects of a HFD on the cognitive and emotional profile of adolescent mice, as well as its effect on the neuronal morphology of the prefrontal cortex. This is a very interesting study, and the contribution of this manuscript is original. The results are interesting, evidencing how HFD can induce consequences in adolescents such as anxiety- and depression-like behavior, and also induce changes in prefrontal neuroplasticity.
However, I do have some comments that the authors should revise prior to its publication. Below I provide some points that hopefully will help the authors to sharpen their manuscript.
Introduction
The important information is already in the manuscript, but it is disseminated, there is a lack of a thread through the text. Authors should focus on the main aim of the present work so I recommend clarifying a few points:
· Authors claim that HFD consumption adversely affects emotional and cognitive functions, but this information is too generic, they should specify more clearly what the consequences are, such as anxiety- and depression-like behavior.
· The relationship between PFC and behavioral outcomes should also be further developed, as well as the relationship between pyramidal neuronal morphology in the mPFC or microglial parameters with anxiety and depression-like behavior.
· Authors indicate that “HFD consumption is prevalent among adolescents” but they should provide some justification for this information.
Results
· Some figures contain excessive repetitive information. In Figure 3, for example, M, N and O present visual information, but the data from these tests have already been presented previously. I recommend reducing and simplifying some panels.
· The statistical test performed is correct to compare data between two groups, but the authors suggest some relationships as “Dendritic atrophy may result in dysregulation of information transfer and processing and thereby induce emotional abnormalities”. Have they performed any correlation, such as Pearson correlation coefficient, to check if there is a direct relationship between anxiety and depression like behavior with pyramidal neuronal morphology in the Mpfc or microglial parameters?
In the neuronal parameters in the mPFC analysis, a sample of 3-4 subjects per group has been analyzed. Can this low sample be representative?
Discussion
Throughout this section, the authors refer to emotional and cognitive impairment to justify the results, but this is a very generic concept that can involve many other impairments. They should specify information related to the cognitive and emotional variables evaluated.
Authors should develop further the reason why the HFD may be leading to anxiety- and depression-like behavior.
The results presented confirm that HFD induces abnormal neuronal morphology, but a more detailed explanation to clarify this effect should be added. What mechanisms can explain that HFD causes these alterations?
Methodology
The authors show the different n analyzed in the behavioral tests, but there are also different samples evaluated in others tests. I recommend adding the n analyzed in the biological tests (Golgi-Cox impregnation and morphometric analyses, Immunohistochemistry, Western blot assay...).
Reviewer 2 Report
In the manuscript entitled “High-fat diet consumption induces neurobehavioral abnormalities and neuronal morphological alterations accompanied by excessive microglial engulfment and reduced BDNF levels in the medial prefrontal cortex in adolescent mice” a timely and challenging working hypothesis is tested. However, a poor defined and weak experimental design leads to inconsistent results, which can unlikely be repeated in other laboratories or extrapolate meaningful conclusions.
Main major concerns:
1. sample size and housing conditions. 4-5 animals/cage are unacceptable. The group-housed mice number is generally 3. This is also important to control variables, i.e. amount of ingested food. Which is the test to define such number of animals?
2. different and invasive behavioural tests performed each day deeply affect mice behaviour leading to a strong bias in the outputs and related analyses.
3. morphological study: to trust IBa1-CD68 please show single images.
4. BDNF decreasing in microglia cells must be demonstrated via IF or by western blot on homogentaes of isolated microglia cells. Microglia are a source of BDNF as well as neurons.
